# Assessing Veterans' Lived Experiences After Exposure to an Autonomous Shuttle

Isabelle Wandenkolk [1,*], Sherrilene Classen [1], Nichole E. Stetten [1], Seung Woo Hwangbo [1] and Kelsea LeBeau [2]

1   Institute for Driving, Activity, Participation, and Technology, Department of Occupational Therapy, University of Florida, Gainesville, FL 32611, USA; sclassen@ufl.edu (S.C.); nichole_stetten@brown.edu (N.E.S.); shwangbo@phhp.ufl.edu (S.W.H.)
2   Veterans Rural Health Resource Center-Gainesville (VRHRC-GNV), North Florida/South Georgia Veterans Health System, U.S. Department of Veterans Affairs, Gainesville, FL 32608, USA; kelsea.lebeau@vumc.org
*   Correspondence: icoppawanden@umass.edu

## Abstract

Transportation is often cited as a significant barrier to healthcare access by Veterans, particularly those from minority groups, who have disabilities, or live in rural areas. Autonomous shuttles (AS) offer a potential solution, yet limited research has explored Veterans' experiences with this technology. This study qualitatively investigated Veterans' lived experiences with AS through focus groups, enrolling participants aged 18+ from Gainesville, The Villages, and Lake Nona, Florida. Via a directed content analysis, six key themes were identified: Perceived Benefits, Safety, Experience with Autonomous Vehicles (AV), AS Experience, AV Adoption, and Perception Change. Among 26 participants (aged 30–85; 77% men; 88% urban residents), prominent themes included Safety (n = 161), Perceived Benefits (n = 153), and AS Experience (n = 118), with predominantly positive counts in all themes except AS Experience. Participants acknowledged safety advantages and multitasking potential of AS over human-operated vehicles while recommending improvements to the shuttle's slow speed, availability and convenience. While the AS ride was positively received overall, some participants noted issues with comfort and braking, emphasizing the need for further technological enhancements. Real-world exposure to AS appeared to influence acceptance positively, offering insights for policymakers and industry stakeholders aiming to optimize AS deployment for mobility-vulnerable Veterans.

**Keywords:** autonomous shuttle; lived experiences; veterans

## 1. Introduction

### 1.1. Transportation Services Available to Veterans

The Veterans Transportation Program (VTP) is a federal program under the Veterans Health Administration, which provides transportation services to enrolled Veterans in need of assistance accessing VA healthcare [1]. Additionally, the CHOICE Act (Veterans Access, Choice, and Accountability Act) and the MISSION Act (Maintaining Internal Systems and Strengthening Integrated Outside Networks Act) aim to expand healthcare options and decrease travel for enrolled Veterans [2]. These Acts address issues of long wait times and limited access to healthcare facilities by allowing enrolled Veterans to seek care from non-VA providers in their local community. However, despite these efforts, enrolled Veterans still encounter transportation barriers when accessing healthcare. Studies have identified challenges faced by enrolled Veterans, including difficulties in transportation to

medical appointments, particularly for military Veterans transitioning back to civilian life and those residing in rural communities [3,4]. Additionally, since both the VTP services and the benefits under the CHOICE and MISSION Acts are designed to address specific healthcare-related needs of enrolled Veterans, they do not cover other essential services like grocery shopping, employment, education, and community support and integration [1].

Moreover, non-enrolled Veterans also cite lack of transportation options as a barrier to accessing healthcare and other essential services. For instance, Veterans with spinal cord injury described that the biggest obstacle to finding and maintaining a job was due to the lack of transportation options [5]. Additionally, Veterans highlighted the need for transportation to health-care appointments and other needs to aid Veterans' military-to-civilian transition [3]. Notably, this resource issue particularly affected elderly Veterans and less affluent Veterans living in rural areas.

While enrollment status significantly impacts eligibility for VA-provided transportation to healthcare appointments, with only about half of all Veterans enrolled in VA healthcare, transportation limitations affect both enrolled and non-enrolled Veterans [6]. Thus, there is a need to explore solutions that address not only healthcare access but also general transportation needs, such as access to employment, education, grocery shopping, and community support services, to contribute to enhanced transportation services for mobility disadvantaged Veterans.

### 1.2. Autonomous Shuttles

Autonomous vehicle (AV) technology represents a revolutionary leap in transportation, characterized by vehicles capable of navigating without direct human intervention. Employing advanced technologies like sensors, cameras, radar and lidar, and artificial intelligence algorithms, AVs interpret their surroundings and make informed decisions to execute optimally safe driving behaviors [7]. The National Highway Traffic Safety Administration (NHTSA) classifies these vehicles from Level 0 to Level 5, with higher levels indicating reduced human control. One notable application of AV technology is the development of the AS [8], a cutting-edge advancement with significant promise for transforming travel.

Autonomous shuttles fall under the category of autonomous ride-sharing services, in which such services facilitate carpooling among travelers with similar routes, optimizing vehicle utilization and reducing road congestion [9]. By acting as first- or last-mile service providers, AS can connect passengers to public transportation hubs or to their final destinations, enhancing mobility options and contributing to a more equitable and sustainable transportation system [10,11]. Notably, AS offer distinct advantages like accessing narrow streets that are not easily served by traditional buses, while also reducing noise and pollution [9]. While the potential benefits of AS technology are substantial [12], challenges and limitations still need to be addressed. Safety concerns, liability issues in accidents involving AS, ethical considerations in decision-making algorithms, and cybersecurity threats are critical aspects that demand attention [13]. Additionally, the current early stages of testing and deployment of AS are characterized by constrained operations, with short fixed schedules adhering to predetermined routes at slow speeds, limiting their flexibility compared to on-demand services [14,15]. Another significant consideration in the deployment of AS is cost. Initial capital investments for AS systems—including vehicle acquisition, infrastructure upgrades, and maintenance—can be substantial, particularly when scaled across healthcare systems or geographically dispersed populations. Moreover, operational costs such as supervision, insurance, and technology updates may affect long-term affordability and sustainability. These financial considerations are especially relevant in the context

of publicly funded services for Veterans, and must be addressed to ensure equitable and practical implementation.

Recognizing the dual nature of benefits and limitations associated with the technology, it is important to assess Veterans' transportation needs and perspectives regarding AS technology, particularly among those facing mobility challenges. This emerging technology holds the potential to bridge current transportation gaps for Veterans and serves as an innovative solution for those lacking reliable options for healthcare appointments and other essential activities.

### 1.3. Qualitative Studies on AS Acceptance

Sociodemographic groups vary in their perceptions of and willingness to adopt AV technology, as commuters prioritize different features and benefits of AVs [16]. For example, a study found that civilian drivers initially held negative perceptions of AS; however, their attitudes improved after riding in the AS, as indicated by post-exposure interviews [17]. Similarly, another study reported increased acceptance of AS among individuals with and without disabilities post-AS exposure in Gainesville, FL [18]. However, one study specifically explored the AS experiences of Veterans [14]. Although Veterans positively responded to their AS experience, the study assessed post-exposure perceptions using a questionnaire. While questionnaires offer valuable insights into user experiences, they may not fully capture the depth of users' perceptions.

Veterans represent a unique population whose experiences and perceptions of AVs may differ significantly from other sociodemographic groups. Due to their military service, Veterans may have heightened awareness of safety and reliability, distinct transportation needs related to healthcare access, and specific psychological or physical conditions (e.g., PTSD, mobility limitations) that influence their engagement with new technologies. These factors may shape their trust, expectations, and acceptance of autonomous shuttles in ways not typically observed in older adults or the general public [19]. Collecting qualitative data to examine the lived experiences of individuals after shuttle exposure offers more detailed insights into their perceptions toward AS, compared to solely relying on survey data [17,20,21]. Given the limited literature focusing specifically on Veterans, this study offers a novel contribution by qualitatively exploring their firsthand experiences with AS and identifying potential opportunities and barriers that may inform future policy, design, and outreach strategies.

### 1.4. Rationale, Significance and Purpose

Veterans often face barriers when it comes to accessing services such as healthcare due to transportation challenges. To mitigate healthcare access disparities, the VA has taken steps to address transportation barriers via the VTP and legislative initiatives. However, the VA's transportation-related efforts are confined to healthcare-related appointments and are exclusively available to enrolled Veterans. Considering the dynamic nature of the transportation landscape, AS is a promising solution to enhance autonomy and accessibility for mobility-vulnerable Veterans. Exploring AS technology as a potential future transportation option for Veterans requires an understanding of the factors that impact the acceptance of such technology among Veterans. Therefore, given the novelty of autonomous technology and the limited exploration of Veterans' lived experiences with AS, this study leverages focus group data to gain deeper insights into Veterans' perceptions, knowledge, and experiences post-AS exposure. This study aims to provide recommendations for VA decision-makers (e.g., VA medical centers), VA transportation stakeholders (e.g., Veterans Transportation Service), and industry partners (e.g., AS manufacturers and service providers), in considering AS deployment for Veterans.

## 2. Materials and Methods

*2.1. Ethics*

The focus group data were obtained from two studies (FY21-22 IRB202101463: PI Classen and FY22-23 IRB202202386: PI Classen), which were approved by the University of Florida's Institutional Review Board, the North Florida/South Georgia Veterans Affairs Human Research Protection Office, and the Office of Rural Health. Participants in the studies provided written consent by signing IRB- and VA-approved documentation, including the Informed Consent Form (ICF) and the Health Insurance Portability and Accountability Act (HIPAA) form, confirming their agreement to participate.

*2.2. Study Design*

This study used focus group data from two studies to assess Veterans' lived experiences post-AS exposure. Study 1 focused on Veterans located in Gainesville, Florida, while Study 2 expanded the geographic scope to include additional sites across Florida, including The Villages and Lake Nona. Although conducted as two separate studies due to differences in study scope and timing, both used the same focus group guide to ensure consistency in data collection.

Both studies used a basic qualitative research design [22]. This design was selected because the primary aim of the study was to understand how Veterans make sense of and describe their direct experiences with AS technology in their own words. The basic qualitative design is particularly well suited for capturing practical perspectives, meanings, and concerns as expressed by participants themselves. Given the applied nature of the research—focused on informing technology deployment and policy decisions for Veteran mobility—the study prioritized actionable insights.

*2.3. Study Population*

All participants were US Veterans aged 18 or older, residing in Florida, and able to read, understand, and verbally respond to the focus group questions in English. English proficiency was determined through self-report to ensure participants could understand the focus group questions and effectively communicate and engage in the group discussions. Veterans were eligible to participate regardless of their combat history or military branch (e.g., Army, Navy, Air Force, Marines), as the study aimed to capture a broad and inclusive range of Veteran experiences. This inclusive approach was intentional given the exploratory nature of the research and the goal of identifying diverse perceptions of AS. Participants who did not have access to a device compatible with the internet and Microsoft Teams were excluded. All participants received compensation for their time.

*2.4. Procedure*

The research team compiled focus group data from the two studies conducted between 2021 and 2023 [23]. In both studies, following the completion of the shuttle ride, participants completed the focus group session. The online focus group sessions were conducted through VA HIPAA-compliant Microsoft Teams (version 1.6.00, Microsoft Corp., Redmond, WA, USA), with each session lasting approximately two hours. A conference room at the Veterans Rural Health Resource Center facility in Gainesville, FL, was reserved for the online focus groups, allowing participants to join from any suitable location. The session began with a period for participants to log on and address any technical issues, followed by the moderator facilitating a discussion around the focus group questions. Payment information was provided at the end of the session, which concluded with a wrap-up.

### 2.5. Data Collection

Although participants were recruited from diverse backgrounds—including varying military service categories, racial and ethnic groups, and disability statuses—these specific demographic details were not consistently collected across all sites. Therefore, this information was not analyzed or reported in this manuscript. The focus group questions aimed to collect participants' views on autonomous ride-sharing services and their experiences with the self-driving shuttle. Focus group questions asked participants about their perceptions, experiences, and opinions about AS technology, covering both pre- and post-ride perceptions, including questions related to comfort, safety, usability, and the potential impact on their personal mobility and daily transportation needs. Each focus group was audio-recorded using Microsoft Teams, with field notes documenting nonverbal cues and ensuring data accuracy by capturing any gaps or inaudible periods of the recording. Afterward, Microsoft Teams automatically transcribed the recordings, and a research assistant reviewed each transcript, cross-referencing with field notes and recordings to ensure completeness and accuracy.

### 2.6. Data Management

The research team took various measures to ensure the security and confidentiality of the data from the two studies. Participants' focus group recordings were securely stored in password-protected systems within a secure VA research office, adhering to VA and University information security policies. All data used for analysis were de-identified to protect the privacy of the participants.

### 2.7. Data Analysis

The focus groups were recorded and transcribed verbatim, and the resulting transcripts were deidentified. A directed content analysis was used to analyze the focus groups, an approach well-suited for topics with existing research or theoretical frameworks that the researcher aims to expand upon [24]. Using a deductive approach, predetermined codes were used as a guide for the analysis, and any text that did not fit within the initial coding scheme was assigned a new code using an inductive approach. A directed approach was specifically chosen to support the application of the conceptual model that informed the development of the Automated Vehicle User Perception Survey [25]. This model provided a structured framework to examine key domains relevant to user perceptions of automated vehicles, including (a) intention to use, (b) perceived ease of use, (c) perceived usefulness, (d) safety, (e) trust and reliability, (f) experience, (g) control and driving efficacy, and (h) external variables (e.g., media, governing authority, social influence, and cost). Applying this model strengthened the theoretical grounding of the analysis and allowed for systematic interpretation of participants' perceptions within an established framework.

During deductive and inductive coding procedures, the constant comparison method was used among two researchers. Having multiple investigators analyze the data can enhance the confirmability and overall trustworthiness of the findings [26,27]. The two coders analyzed the data using a structured approach consisting of three main phases of content analysis: (1) Preparation, (2) Organization, and (3) Reporting [28]. During the preparation phase, the coders immersed themselves in the data by reading the transcripts multiple times [24]. In the organization phase, they read the data line by line and categorized the data according to the predetermined coding scheme or formed new codes based on patterns observed in the data.

Once all the themes and subthemes were identified, the qualitative data were further classified into positive, negative, and neutral categories. This additional level of stratification was performed to enrich the interpretation of findings by capturing not only the

content of participant responses but also the tone and directionality of their perspectives. By distinguishing between positive, negative, and neutral sentiments, the research team was better able to surface patterns of endorsement or resistance, pinpoint specific pain points or areas of satisfaction, and identify where participants held ambivalent or mixed views. This level of detail supports a more actionable understanding of stakeholder feedback, particularly when designing interventions, improving services, or informing policy [29].

Specifically, the data were categorized as positive if responses expressed favorable opinions, satisfaction, agreement, or endorsement of a given concept, topic, or experience. Positive responses often highlighted benefits, strengths, or positive attributes associated with a theme/subtheme. Responses categorized as negative expressed unfavorable opinions, dissatisfaction, disagreement, or concerns regarding a particular concept, topic, or experience. Negative responses highlighted drawbacks, challenges, or negative aspects associated with a theme/subtheme. Responses categorized as neutral did not strongly convey a positive or negative perception. Neutral responses provided information, observations, or opinions without expressing a clear preference or inclination toward the positive or negative aspects of a theme/subtheme.

In the organizational phase, investigator triangulation was used to enhance trustworthiness [27,30], particularly pertaining to the credibility of the findings. This strategy was used to ensure the plausibility of information from participants' original data and accurate interpretation of the participants' original views. Specifically, two researchers regularly met for thorough discussions encompassing the entire research process, which involved coding, exploration of the themes and subthemes, and interpretations of contexts and rationales. Coders documented detailed notes of the coding and decision-making processes to ensure neutrality and unbiased results. Using the constant comparison method [31], these meetings persisted until a consensus was reached. Any disagreements were resolved with the help of a third team member experienced in content analysis. The final reporting phase involved the presentation and interpretation of the resulting findings.

## 3. Results

The focus groups included participants with ages ranging from 30 to 85, comprising 77% men, and 88% living in urban areas. Participants in The Villages had a mean age of 72 years, while those in Lake Nona presented a younger mean age of 42 years, and participants in Gainesville had a mean age of 58 years. Gender distribution remained consistent across locations, with about 63% being male. A total of nine focus groups were conducted across both studies: four in Gainesville, three in Lake None, and two in The Villages, with 26 Veterans participating in total.

Table 1 illustrates the qualitative themes, subthemes, and their respective operational definitions. A directed content analysis revealed six major themes and seven subthemes. The six major themes included the following: Perceived Benefits, Safety, Experience with AV, AS Experience, AV Adoption, and Perception Change. Perceived Benefits encompassed three subthemes: Perceived ease of use, Availability, and Accessibility. Safety comprised one subtheme (i.e., Trust and reliability), AS Experience included two subthemes (i.e., Comfort and Speed), and AV Adoption had one subtheme (i.e., External variables). Experience with AV and Perception Change themes did not generate any subthemes.

**Table 1.** Qualitative themes and subthemes operational definitions.

| Themes Subthemes | Operational Definitions of Themes and Subthemes |
|---|---|
| Perceived Benefits | Individual's perception of the usefulness of AVs, including factors such as the perceived value, benefits, and advantages of using AVs over traditional vehicles. |
| Perceived ease of use | Individual's perception of the effort required to use AVs, including perceived complexity, ease of learning, and the ease of interacting with the technology (user-friendly). |
| Availability | Availability of AVs in the local area or access to AV services/providers. Adequacy of infrastructure to support AV usage, including availability of charging stations and support systems for maintenance and repairs. |
| Accessibility | The consideration of diverse user needs, including individuals with disabilities, elderly users, or users with varying technological literacy, and the provision of accessible features or accommodations in AVs. |
| Safety | Individual's perception of the safety of AVs, including perceived risks, hazards, and potential accidents associated with AVs. |
| Trust and reliability | Individual's perceptions of the trustworthiness and dependability of AVs, including confidence in the technology's ability to navigate safely and effectively in various driving scenarios, and vehicle performance reliability. |
| Experience with AV | Individual's actual experience with AVs and/or AV technology. It includes factors such as the individual's past interactions with AVs and/or AV technology and the feedback received from other users. |
| AS Experience | Individual's experiences specifically related to using the study's AS. It includes aspects such as the ease of boarding and disembarking, the overall efficiency of the shuttle system, and any notable positive or negative experiences encountered during their shuttle rides. |
| Comfort | Individual's perceptions of comfort while using the AS, including feelings of physical comfort (e.g., seat comfort, vehicle ergonomics) as well as psychological comfort (e.g., feeling safe, relaxed, or confident) during the shuttle ride. |
| Speed | Individual's experiences with the AS's speed in relation to their expectations or preferences, including aspects related to the vehicle's acceleration, deceleration, and overall speed during the ride. |
| AV Adoption | Individual's inclination or readiness to utilize AVs in the future. It encompasses their expressed motivations, barriers, and factors influencing their intention to use autonomous vehicles for their transportation needs. |
| External variables | External factors that may influence the adoption of AVs, including media coverage, governing authority regulations, social influence, and cost. |
| Perception Change | Perception change refers to the shift in individuals' beliefs, attitudes, or perspectives related to AV technology as a result of their exposure, experience, and/or knowledge acquisition. |

Note. Underlined items indicate overarching themes; non-underlined items represent associated subthemes.

Tables 2 and 3 provide a comprehensive overview of the frequency counts for each theme and subtheme, categorized by location, and further distinguished by positive, negative, and neutral counts. This breakdown offers a clear visualization of the prominence of specific themes and subthemes within the focus group discussions. In the analysis of participant responses, a subset was classified as neutral because they did not strongly convey a positive or negative perception, thus not fitting into either positive or negative classifications. These neutral responses consisted of information, observations, or opinions without expressing a clear perception toward the positive or negative aspects of a theme/subtheme. As a result, participants' responses classified as neutral are not discussed in the results section to maintain focus on more definitive perceptions or experiences from the participants. This approach aims to provide clarity and conciseness in presenting the most salient findings while avoiding potential dilution of the results with less decisive or ambiguous responses.

**Table 2.** Frequency counts of themes, categorized by positive, negative, and neutral, across study locations.

| Theme | Frequency Counts | | | |
| --- | --- | --- | --- | --- |
| | Gainesville (n = 10) | Lake Nona (n = 8) | The Villages (n = 8) | Total (N = 26) |
| Perceived Benefits | 70 | 49 | 34 | 153 |
|   Positive | 47 | 39 | 23 | 109 |
|   Negative | 15 | 6 | 7 | 28 |
|   Neutral | 8 | 4 | 4 | 16 |
| Safety | 66 | 64 | 31 | 161 |
|   Positive | 32 | 29 | 9 | 70 |
|   Negative | 19 | 16 | 18 | 53 |
|   Neutral | 15 | 19 | 4 | 38 |
| Experience with AV | 17 | 6 | 7 | 30 |
|   Positive | 8 | 2 | 3 | 13 |
|   Negative | 0 | 0 | 0 | 0 |
|   Neutral | 9 | 4 | 4 | 17 |
| AS Experience | 47 | 45 | 26 | 118 |
|   Positive | 17 | 17 | 1 | 35 |
|   Negative | 25 | 22 | 19 | 66 |
|   Neutral | 5 | 6 | 6 | 17 |
| AV Adoption | 44 | 19 | 16 | 79 |
|   Positive | 37 | 12 | 7 | 56 |
|   Negative | 1 | 4 | 8 | 13 |
|   Neutral | 6 | 3 | 1 | 10 |
| Perception Change | 10 | 6 | 3 | 19 |
|   Positive | 7 | 3 | 0 | 10 |
|   Negative | 1 | 1 | 1 | 3 |
|   Neutral | 2 | 2 | 2 | 6 |

**Table 3.** Subtheme frequency counts categorized by positive, negative, and neutral counts by study location.

| Theme/Subtheme | Frequency Counts | | | |
| --- | --- | --- | --- | --- |
| | Gainesville (n = 10) | Lake Nona (n = 8) | The Villages (n = 8) | Total (N = 26) |
| Perceived Benefits | | | | |
|   Perceived ease of use | 14 | 1 | 7 | 22 |
|     Positive | 11 | 0 | 1 | 12 |
|     Negative | 2 | 0 | 3 | 5 |
|     Neutral | 1 | 1 | 3 | 5 |
|   Availability | 29 | 8 | 4 | 41 |
|     Positive | 12 | 2 | 1 | 15 |
|     Negative | 12 | 4 | 2 | 18 |
|     Neutral | 5 | 2 | 1 | 8 |
|   Accessibility | 11 | 15 | 4 | 30 |
|     Positive | 8 | 13 | 2 | 23 |
|     Negative | 1 | 2 | 2 | 5 |
|     Neutral | 2 | 0 | 0 | 2 |
| Safety | | | | |
|   Trust and reliability | 43 | 28 | 5 | 76 |
|     Positive | 23 | 12 | 0 | 35 |
|     Negative | 11 | 9 | 5 | 25 |
|     Neutral | 9 | 7 | 0 | 16 |

**Table 3.** *Cont.*

| Theme/Subtheme | Frequency Counts | | | |
|---|---|---|---|---|
| | Gainesville (n = 10) | Lake Nona (n = 8) | The Villages (n = 8) | Total (N = 26) |
| AS Experience | | | | |
| Comfort | 6 | 11 | 8 | 25 |
| Positive | 3 | 3 | 0 | 6 |
| Negative | 3 | 7 | 8 | 18 |
| Neutral | 0 | 1 | 0 | 1 |
| Speed | 23 | 8 | 9 | 40 |
| Positive | 0 | 2 | 0 | 2 |
| Negative | 19 | 6 | 8 | 33 |
| Neutral | 4 | 0 | 1 | 5 |
| AV Adoption | | | | |
| External variables | 14 | 9 | 5 | 28 |
| Positive | 10 | 4 | 2 | 16 |
| Negative | 1 | 2 | 3 | 6 |
| Neutral | 3 | 3 | 0 | 6 |

Frequency counts were determined by the number of times a theme/subtheme was mentioned by participants. The combined frequency counts across all study locations revealed that Safety (n = 161), Perceived Benefits (n = 153), and AS Experience (n = 118) emerged as the top three themes. The themes AV Adoption (n = 79), Experience with AV (n = 30), and Perception Change (n = 19) followed closely in frequencies. Examining the breakdown by location, a consistent pattern emerged, with Safety, Perceived Benefits, and AS Experience occupying the top three spots, albeit with some variation. Notably, Gainesville and The Villages diverged from the overall pattern, with Perceived Benefits ranking highest in frequency, followed by Safety and AS Experience. For the theme Safety across locations, Gainesville and Lake Nona displayed more positive frequency counts than negative, while The Villages exhibited the opposite trend. Conversely, for Perceived Benefits, positive counts dominated across all locations, with variations in subtheme priorities. Within Perceived Benefits positive frequency counts, the most frequent subtheme in Gainesville was Availability, while Accessibility took precedence in Lake Nona and The Villages. Negative frequency counts for Perceived Benefits were primarily attributed to Availability in Gainesville and Lake Nona, and Perceived Ease of Use in The Villages. Within the AS Experience theme, all locations reported more negative than positive frequency counts. In Gainesville, negative counts were concentrated in the Speed subtheme, while Lake Nona and The Villages demonstrated a more balanced distribution between Comfort and Speed. The theme AV Adoption displayed a predominantly positive trend in Gainesville and Lake Nona, whereas The Villages presented more negative than positive frequency counts. Experience with AV yielded solely positive frequency counts across all locations. Finally, Perception Change exhibited more positive than negative counts in Gainesville and Lake Nona, with The Villages reporting a single negative count and no positive counts.

The themes that emerged from the coding among Gainesville, The Villages, and Lake Nona focus groups are synopsized as follows.

### 3.1. Perceived Benefits

Across all study locations, participants consistently recognized the need for AS services, particularly in areas around VA hospitals and for the elderly population.

- Positive Perception of Availability: "They need it [AS] in a lot of the places like Gainesville, Ocala, especially around the Veterans hospitals (ID: 1070)."

- Positive Perception of Accessibility: "A lot of the more elderly are on meds. And the meds impair their driving. It will cause them to fall asleep or whatever. Unfortunately, maybe have a heart attack behind the wheel, and self-driving, you could say it could cause less accidents (ID: 4)."

They cited various benefits, such as the ability to multitask during the ride and the potential to enhance traffic flow and road safety by minimizing human errors and road rage, factors often linked to accidents.

- Positive Perception: "Well, one thing it saves energy. If I'm going to a meeting or something that gives me time to look over my papers, you know? You don't have to deal with traffic, you don't have to actually drive, you could be on your phone. There's a lot of things you could be doing while you are riding it (ID: 1044)."
- Positive Perception: "You take the human equation out, then the cars would go at a steady speed. You wouldn't have these crazies weaving in and out, which slows traffic down (ID: 7)."

Post-AS experience, participants regarded it as an effortless mode of transportation. For example,

- Positive Perception of Ease of Use: "I don't think there's anything I didn't like about the shuttle. It was easy, it was comfortable (ID: 1070)."

Conversely, concerns were raised regarding convenience in terms of availability (i.e., reaching desired destinations like the VA hospital) and timeliness (i.e., operating times and days, punctuality of services, and potential tardiness to appointments).

Negative Perception: "Is it convenience? Is it gonna get me there faster? I wouldn't go out of my way to use it and I wouldn't be late to use it (ID: 1042)."

### 3.2. Safety

Aligned with Perceived Benefits theme, participants consistently conveyed a strong sense of trust and confidence in the safety and reliability of the AS, often emphasizing it as safer and more reliable than human-operated vehicles.

Positive Perception: "Statistically, they're safer—they take the human equation out, so statistically, they're safer than humans behind the wheel (ID: 4)."

Positive Perception of Trust and reliability: "Honestly, I think that will reduce a lot of accidents because then there won't be all these drivers out there crossing people out (ID: 134)."

Negative narratives, however, revolved around concerns such as potential hacking and issues with the AS sensors, which, if overly cautious, could pose risks around other drivers and in high-density traffic. While participants expressed trust in the shuttle's safety standards and cautious operations, the need for ongoing supervision and further improvements for the seamless integration of AS services with other road users was highlighted.

- Negative Perception: "Well, these automated vehicles, they'd be prone to hacking. I mean, they're all computerized, right? What would stop a hacker from hacking these? (ID: 7)."
- Negative Perception: "It was super cautious because it would like come to a complete stop just because the bicycle came near (ID: 1042)."
- Negative Perception of Trust and reliability: "I'm a little wary of vehicles that auto brake for you because they do it too suddenly. If it can gradually break, instead of having like a stop on you, then it would be better (ID: 104)."

### 3.3. Experience with AV

Participants shared insights into their past encounters with AVs. Some participants had prior experiences with AVs, ranging from airport shuttles to Teslas or Advanced Driver Assistance Systems like adaptive cruise control. Notably, all experiences with the AS were characterized as positive; no narrative examples for Experience with AV were identified as negative.

- Positive Perception: "I don't remember which airport, but one of the airports I was at about a year ago had an autonomous shuttle. It was a good experience (ID: 1038)."
- Positive Perception: "I have a brother in Tampa who has a Tesla. It drove us home perfectly (ID: 1060)."
- Positive Perception: "It was like when you got your cruise control, and all of a sudden you come close to this car, and it slows down. And I'm like, oh, this is really cool (ID: 8)."

### 3.4. Autonomous Shuttle Experience

Notably, participants in Gainesville and Lake Nona shared positive experiences, emphasizing comfort during their rides in the AS.

- Positive Perception of Comfort: "You could fit probably 6 people very comfortably in it, plus a few standing locations too. It was very clean. Smell very nice, very user friendly. I like it. (ID: 1010)."

Interestingly, The Villages participants expressed a positive and enjoyable experience riding the shuttle, but comfort was not mentioned as a contributing factor. Conversely, negative experiences were voiced across all sites, despite some participants overall finding the ride positive and comfortable. Common concerns included the hard bus-like seats, harsh braking, and slow speed, which raised potential safety considerations.

- Negative Perception of Comfort: "More comfortable, like the seats. So, if I could suggest like adding a cushion or something especially, I don't know for probably older people (ID: 108)."
- Negative Perception of Comfort: "They need to improve the braking system, that hurt my back. Every time you stop, it was painful (ID: 5)."
- Negative Perception of Speed: "Going so slow it might actually cause an accident because a lot of times impaired or just drivers that don't pay attention will be expecting to continue at a standard flow, and the autonomous shuttle seems to be a little slower than that (ID: 1014)."

Participants consistently expressed the need for improved comfort, especially during longer trips. Some participants felt that the potential of the shuttle was not realized due to operator intervention, although there were contradictory views, with some expressing satisfaction with having a staff person in the AS to optimize safety.

- Negative Perception: "I thought it'd be—I didn't know she [operator assistant] was going to have to hit the go button each time. I thought it would be fully autonomous and it wasn't (ID: 3)."
- Positive Perception: "I did like that we had a safety person on there. I think they should have somebody sit on there just in case, even if they advance the technology for safety reasons (ID: 104)."

### 3.5. AV Adoption

Positive narratives from participants in Gainesville, Lake Nona, and The Villages underscored an overall enthusiasm for new technology, with statements expressing a willingness to use AS regularly.

- Positive Perception: "I would absolutely use it [autonomous shuttle] regularly (ID: 1014)."
- Positive Perception: "I'm actually looking forward where I won't have to own a car anymore. Just go out there and it drives me everywhere (ID: 134)."
- Participants expressed varied intentions for using the AS, with some utilizing it for VA appointments and others for leisure activities.
- Positive Perception: "Take me to my Vet appointment. I'll take the shuttle (ID: 6)."
- Positive Perception: "I would say more for like leisure you know, if you're going out to have a good time, you know, have some dinner to have some drinks (ID: 116)."

External variables played a pivotal role in participants' considerations, emphasizing potential benefits for Veterans, including alleviating transportation challenges and reducing associated expenses. Some viewed AS as the "wave of the future," anticipating improved safety and a more relaxed travel experience.

- Positive Perception of External Variables: "I think it will help a lot of Veterans who don't have forms of transportation. It can help them get around to wherever they need to go. You know they need to get to an appointment or something like that. Umm, I think it'll help them without them having to like spend a lot of money (ID: 116)."
- Positive Perception of External Variables: "Well, I just think that they're the wave of the future and because it'd be so much safer, and I think driving back and forth to Indiana every year, it would be so much nicer not having to fight the traffic (ID: 2)."

For negative narratives, concerns were voiced about incidents or complaints post-implementation and a preference for the freedom and control of self-driving, particularly among Lake Nona participants.

- Negative Perception of External Variables: "Like after these vehicles are put into service, you have a number of large complaints or incidents that occur. That would probably be the only thing I would certainly listen to (ID: 1006)."
- Negative Perception: "I prefer to drive myself because I can control the speed, set the AC to my liking, and play the music I want (ID: 102)."

Interestingly, skepticism emerged, specifically among participants in The Villages, rooted in developmental challenges and the need for improvements in the technology's smoothness before widespread adoption. Specific concerns included discomfort in long-distance travel, potential high costs, and safety issues reported in the media.

- Negative Perception: "It's not smooth yet, so I think it just needs some improvements before it's really ready to say I would use it regularly (ID: 8)."
- Negative Perception: "To the VA hospital? No, that's too far. My back's screwed up. I couldn't really handle it for that long (ID: 3)."
- Negative Perception of External Variables: "It's going to be a while because who wants to pay $10,000 for a new battery and an electric vehicle? (ID: 3)."
- Negative Perception of External Variables: "I certainly know about their accidents because they appear on Google. Kill somebody running into a wall or whatever (ID: 1)."

*3.6. Perception Change*

Participants discussed how their perceptions might have changed before and after riding in the AS. Positive narratives, specifically from participants in Gainesville and Lake Nona, revealed an improvement in perceptions, with increased excitement, positive shifts in attitudes, and a growing comfort and understanding of the technology.

- Positive Perception: "Honestly, for the better, you know, gives me hope (ID: 114)."

- Positive Perception: "I would say yes. So, after taking a ride in it, I got to understand a lot more and I just became more comfortable with the whole idea and the whole concept (ID: 116)."

For some participants already favorably inclined towards AVs, the AS experience reinforced or validated their existing beliefs.

- Positive Perception: "I'm leaning more for it then against it. I was already more for it. I'm even more for it now (ID: 1044)."

Notably, The Villages participants did not show a positive attitude shift. Negative narratives across all sites primarily centered on unmet expectations, expressing disappointment in terms of distance, smoothness, and overall experience, underscoring the need for ongoing advancements in the technology.

- Negative Perception: "I expected a lot more (ID: 6)."
- Negative Perception: "It was different than I had in my head. I thought it would be like a little longer, a little smoother (ID: 102)."
- Negative Perception: "I thought it was a lot further. It seems like we got a long way to go (ID: 1042)."

In summary, the results indicate that while participants acknowledged the potential benefits of AS services, concerns related to convenience, safety, and technological advancements are key considerations for widespread acceptance. To address convenience-related concerns, it is essential to optimize routes and schedules. Safety concerns, particularly regarding hacking, AS speed and the sensitivity of AS sensors, highlight the need for continuous enhancement of security measures and technology refinement. Ongoing technological advancements, especially in addressing smoothness and performance issues, are essential for aligning AS with user expectations.

*3.7. Co-Occuring Thematic Constructs*

Table 4 presents the count and percentage of quotes containing co-occurring thematic constructs among all study participants, encompassing positive, negative, and neutral quotes. The columns represent different themes and subthemes, determining the count denominators. For the theme Perceived Benefits, mentioned a total of 153 times by participants, Safety co-occurred at the highest percentage (22%). The subtheme Perceived Ease of Use co-occurred (41%) with Perceived Benefits. This means that the quotes encompassed information specifically regarding benefits related to Perceived Ease of Use and general Perceived Benefits that did not fit into a subtheme. Perceived Ease of Use also co-occurred the most with AS Experience (32%). The subtheme Availability demonstrated a prominent co-occurrence with its theme, Perceived Benefits (32%), and AV Adoption (27%). The Accessibility subtheme had the highest co-occurrence percentage with Safety (40%).

For the theme Safety and subtheme Trust and reliability, AS Experience co-occurred at the highest percentage (27% and 24%, respectively). Interestingly, for the theme AS Experience (37%) and subtheme Speed (58%), Safety co-occurred at the highest percentage. However, AS Experience's other subtheme, Comfort, co-occurred with other themes, including Perceived Benefits and AV Adoption (both at 12%). For Experience with AV (23%) and Perception Change (32%), Safety exhibited the highest co-occurrence percentage. Perception Change also co-occurred with Safety's subtheme Trust and reliability (32%). Lastly, the theme AV Adoption and its subtheme External Variables, had the highest co-occurrence percentage with Perceived Benefits (33% and 43%, respectively). Figure 1 visually illustrates the most prominent co-occurring thematic relationships identified in the data. Thematic nodes represent themes and subthemes, while directional arrows and percentages denote the strongest co-occurrence patterns.

**Table 4.** Percentage and number of quotes that contain co-occurring thematic constructs for all study participants.

| Themes/ Subthemes | PB | PEU | AVA | ACC | Safety | T&R | E-AV | AS-E | Comfort | Speed | AV-A | EV | PC |
|---|---|---|---|---|---|---|---|---|---|---|---|---|---|
| PB | - | 40.9% 9/22 | 31.7% 13/41 | 23.3% 7/30 | 21.1% 34/161 | 14.5% 11/76 | 13.3% 4/30 | 14.4% 17/118 | 12.0% 3/25 | 12.5% 5/40 | 32.9% 26/79 | 42.9% 12/28 | 0.0% 0/19 |
| PEU | 5.9% 9/153 | - | 12.2% 5/41 | 10.0% 3/30 | 1.9% 3/161 | 2.6% 2/76 | 0.0% 0/30 | 5.9% 7/118 | 8.0% 2/25 | 5.0% 2/40 | 6.3% 5/79 | 3.6% 1/28 | 0.0% 0/19 |
| AVA | 8.5% 13/153 | 22.7% 5/22 | - | 13.3% 4/30 | 2.5% 4/161 | 2.6% 2/76 | 3.3% 1/30 | 5.1% 6/118 | 4.0% 1/25 | 2.5% 1/40 | 13.9% 11/79 | 21.4% 6/28 | 0.0% 0/19 |
| ACC | 4.6% 7/153 | 13.6% 3/22 | 9.8% 4/41 | - | 7.5% 12/161 | 4.0% 3/76 | 3.3% 1/30 | 1.7% 2/118 | 4.0% 1/25 | 0.0% 0/40 | 8.9% 7/79 | 14.3% 4/28 | 0.0% 0/19 |
| Safety | 22.2% 34/153 | 13.6% 3/22 | 9.76% 4/41 | 40.0% 12/30 | - | 1.3% 1/76 | 23.3% 7/30 | 37.3% 44/118 | 4.0% 1/25 | 57.5% 23/40 | 15.2% 12/79 | 17.9% 5/28 | 31.6% 6/19 |
| T&R | 7.2% 11/153 | 9.1% 2/22 | 4.9% 2/41 | 10.0% 3/30 | 0.6% 1/161 | - | 16.7% 5/30 | 15.3% 18/118 | 0.0% 0/25 | 17.5% 7/40 | 7.6% 6/79 | 3.6% 1/28 | 31.6% 6/19 |
| E-AV | 2.6% 4/153 | 0.0% 0/22 | 2.4% 1/41 | 3.3% 1/30 | 4.4% 7/161 | 6.6% 5/76 | - | 0.9% 1/118 | 4.0% 1/25 | 0.0% 0/40 | 6.3% 5/79 | 7.2% 2/28 | 0.0% 0/19 |
| AS-E | 11.1% 17/153 | 31.8% 7/22 | 14.6% 6/41 | 6.7% 2/30 | 27.3% 44/161 | 23.7% 18/76 | 3.3% 1/30 | - | 24.0% 6/25 | 17.5% 7/40 | 11.4% 9/79 | 7.1% 2/28 | 5.3% 1/19 |
| Comfort | 2.0% 3/153 | 9.09% 2/22 | 2.4% 1/41 | 3.3% 1/30 | 2.5% 4/161 | 0.0% 0/76 | 3.3% 1/30 | 5.1% 6/118 | - | 12.5% 5/40 | 3.8% 3/79 | 0.0% 0/28 | 0.0% 0/19 |
| Speed | 3.3% 5/153 | 9.1% 2/22 | 2.4% 1/41 | 0.0% 0/30 | 14.3% 23/161 | 9.2% 7/76 | 0.0% 0/30 | 5.9% 7/118 | 20.0% 5/25 | - | 6.3% 5/79 | 0.0% 0/28 | 0.0% 0/19 |
| AV-A | 17.0% 26/153 | 22.7% 5/22 | 26.8% 11/41 | 23.3% 7/30 | 7.5% 12/161 | 7.9% 6/76 | 16.7% 5/30 | 7.6% 9/118 | 12.0% 3/25 | 12.5% 5/40 | - | 0.0% 0/28 | 5.3% 1/19 |
| EV | 7.9% 12/153 | 4.6% 1/22 | 14.6% 6/41 | 13.3% 4/30 | 3.1% 5/161 | 1.3% 1/76 | 6.7% 2/30 | 1.7% 2/118 | 0.0% 0/25 | 0.0% 0/40 | 0.0% 0/79 | - | 0.0% 0/19 |
| PC | 0.0% 0/153 | 0.0% 0/22 | 0.0% 0/41 | 0.0% 0/30 | 3.7% 6/161 | 7.9% 6/76 | 0.0% 0/30 | 0.9% 1/118 | 0.0% 0/25 | 0.0% 0/40 | 1.3% 1/79 | 0.0% 0/28 | - |

Note. PB = perceived benefits; PEU = perceived ease of use; AVA = availability; ACC = Accessibility; T&R = trust and reliability; E-AV = experience with AV; AS-E = AS Experience; AV-A = AV adoption; EV = external variables; PC = perception change. Underlined items indicate overarching themes; non-underlined items represent associated subthemes.

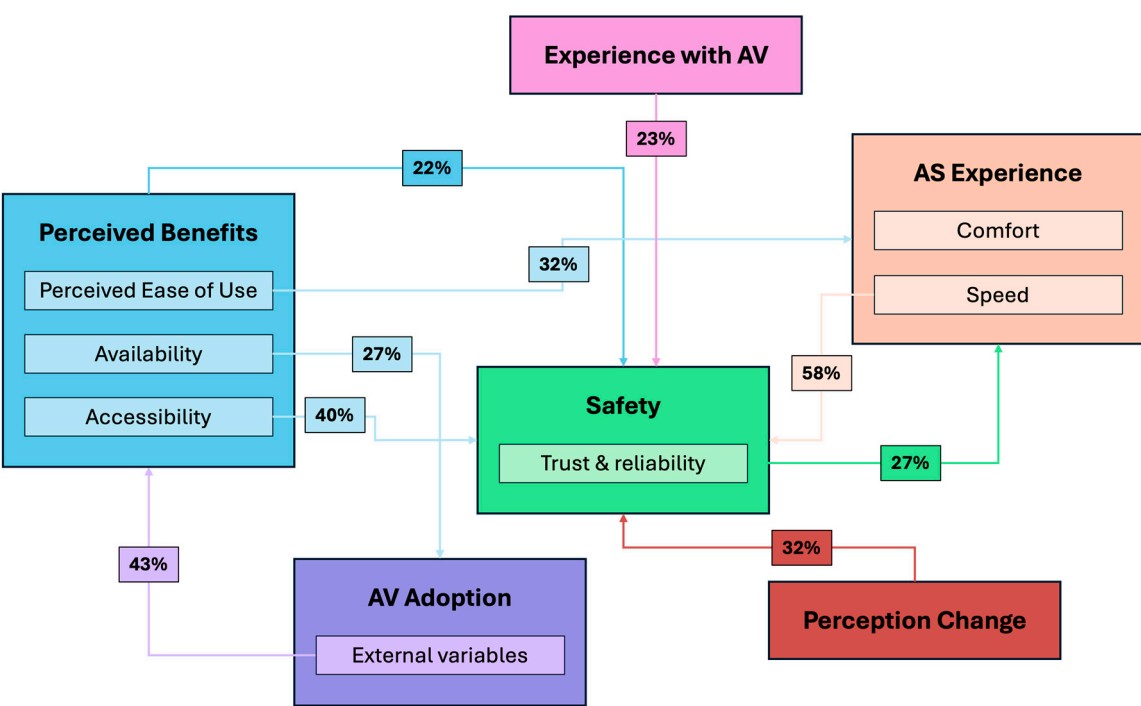

**Figure 1.** Thematic co-occurrence diagram illustrating the most prominent relationships across themes and subthemes.

In summary, the strong co-occurrence between Perceived Benefits and Safety suggests that safety may be an important factor in shaping perceptions regarding the perceived benefits of AS. Participants prioritize convenience and user-friendliness, evident in the association between Perceived Ease of Use and AS Experience. Availability of service routes influences perceptions of AV adoption, as evidenced by the notable co-occurrence between Availability and AV Adoption. Specifically, broader coverage positively influences adoption by enhancing convenience and availability, while limited coverage hinders adoption. Safety considerations, particularly regarding speed, influence participants' experiences with AS, as highlighted in the strong co-occurrence between AS Experience and Safety. For instance, participants who felt confident in the AS's ability to maintain safe speeds and respond effectively to potential hazards characterized their AS experience positively. Conversely, instances where participants perceived the AS as compromising safety, whether due to concerns about the slow speed or abrupt braking, impacted their AS experience negatively. Overall, these findings offer an initial understanding of how participants perceptions of AS technology may be connected based on various themes as identified above.

## 4. Discussion

This study assessed the lived experiences of Veterans (N = 26) after exposure to the AS, using existing focus group data from two studies to gain deeper insights into the participants' perceptions, knowledge, and experiences with AS. The total sample size and number of focus groups align with accepted qualitative research practices, which recommend 3–5 focus groups to reach thematic saturation [32,33]. Given the exploratory nature of this study and the relative homogeneity of the sample, the number of participants was sufficient to support meaningful thematic analysis. The focus group demographics indicate that participants had a mean age of 57 years with 77% male Veterans, aligned with the demographics of the broader Veteran population in the United States [34]. The older age group in The Villages [35] and younger age group in Lake Nona [36] reflect the demographics of these communities.

### 4.1. Safety

Participants' most common theme was Safety, which illuminated the intricate interplay between initial concerns and the development of trust and confidence in AS Safety. Positive narratives highlighted the perceived safety superiority of AS over human-operated vehicles, aligning with findings from a study in which participants expected reduced crashes due to human-errors [17]. Negative narratives centered on concerns about potential hacking and issues with AS sensors. Participants also voiced apprehension about the AS's slow speed and overly cautious behavior, citing potential risks in high-density traffic situations and with other drivers. In contrast to this finding, a recent study reported that participants viewed the slow speed of the AS as enhancing their safety perception, making potential riders feel safer [37]. However, the study by Etminani-Ghasrodashti et al. did not expose participants to an AS; data were collected solely via focus groups. This distinction raises questions about the transferability of findings from a focus group-only study to real-world experiences with AS. Further research, particularly involving actual exposure to AS experiences, is crucial for understanding the real-world impact of slow AS speed on participants' safety perceptions.

Variations in Veterans' safety perceptions were observed across study locations, with participants in Gainesville and Lake Nona having more positive perceptions, while those in The Villages, known for its older population, showed more negative perceptions. On the contrary, a study found that older adults, after experiencing a ride in the AS, felt extremely safe and expressed a willingness to use it [38]. While demographic factors, such as age, may have contributed to the variations in safety perceptions among Veterans across different locations, ongoing research is needed to establish definitive conclusions between age and safety perceptions.

### 4.2. Perceived Benefits

For Perceived Benefits, the second most common theme, the prevalence of positive frequency counts across locations indicates an overall favorable perception of the potential advantages of AS services among Veterans. One positive aspect highlighted by participants across all locations is the perceived ability to multitask during the ride, suggesting that AS services may enhance the overall efficiency and utility of transportation for users. This aligns with findings from Salonen and Haavisto, where participants reported experiencing stress relief and the freedom from complex driving-related tasks while riding the AS [17]. Furthermore, participants acknowledged the potential benefits of AS in improving traffic flow and road safety by minimizing human errors and road rage. These positive perceived benefits align with Veterans' safety perspectives in this study and with existing literature emphasizing the societal advantages, such as reduced accidents and improved traffic flow, of AS in enhancing mobility and safety [7,17].

Negative frequency counts for Perceived Benefits, while present, are relatively minor and are linked to specific concerns regarding the convenience aspects of AS services, particularly related to availability and timeliness. These concerns, such as reaching desired destinations like the VA hospital and ensuring punctuality of services, underscore practical considerations that may influence Veterans' acceptance of AS technology. Participants emphasized the potential for broader community benefits if the shuttle routes were expanded, resonating with the notion of addressing transportation gaps, especially in underserved areas. Notably, participants highlighted the positive impact of expanded routes on accessibility, particularly for vulnerable populations such as the elderly and disabled, underscoring the potential for societal inclusivity. These findings align with other studies reporting that AS route, schedule, and accessibility contribute to the acceptance of AS services [17,37]. Addressing these concerns through effective service planning and commu-

nication strategies could contribute to the successful integration of AS into transportation systems. The variations in subtheme priorities across locations, with Availability being more emphasized in Gainesville, Accessibility in Lake Nona, and The Villages focusing on Perceived Ease of Use, suggest that local contexts and user demographics may play a role in shaping preferences and priorities.

### 4.3. Autonomous Shutlle Experience

Autonomous Shuttle Experience, the third most common theme, encompassed participants' reflections on their firsthand encounters with the AS. While some participants reported positive and comfortable experiences during their rides, concerns were voiced about certain aspects of the experience. Common concerns included the perceived discomfort of hard, bus-like seats, harsh braking, and slow speed, raising potential safety considerations. These results were consistent with the literature on negative responses regarding Speed, as participants reported that the current slow speed of the AS could prevent them from using it [17]. Participants consistently expressed the need for improved comfort, especially during longer trips, indicating that addressing these issues is key for enhancing the overall satisfaction of Veterans. These findings point to optimizing both the physical comfort of the shuttle and its speed to ensure a positive and satisfactory user experience. Some participants felt that the potential of the shuttle was not realized due to operator intervention, although there were contradictory perceptions, with some expressing satisfaction with having a staff person in the AS for safety reasons. Previous research suggests that the presence of a safety operator onboard can increase the sense of safety in passengers [17,37]. The impact of the safety operator on AS experience and safety perception of the AS requires further investigation for conclusive insights.

### 4.4. AV Adoption

For AV Adoption, the fourth most common theme, positive narratives reflect an overarching enthusiasm for new technology, particularly AS. The expressed willingness to use AS regularly suggests a positive disposition toward incorporating AS into participants' transportation routines. The varied intentions for using AS, such as for VA appointments or leisure activities, indicate potential versatility in the adoption of AS technology to meet diverse needs. External variables, such as cost, media coverage and portrayal, and regulations, emerge as influential factors shaping participants' considerations regarding AV adoption. Participants mentioned potential benefits for Veterans, envisioning AS as a solution to transportation challenges and a means to reduce associated expenses. The perception of AS as the "wave of the future" indicates an optimistic outlook, with expectations of improved safety and a more relaxed travel experience. These findings related to the participants' willingness to use AS, associated with cost-reduction, and an increase in safety and travel experience perceptions, align with previous research [17,37].

Conversely, negative narratives regarding AV adoption demonstrated hesitancy or unwillingness among some participants, particularly in The Villages. Notably, this unwillingness to use the AS, especially among participants in The Villages, was linked to AS technological challenges, such as slow speed and sensitive sensors, highlighting the necessity for smoother technology and improvements before widespread adoption. Interestingly, The Villages was the only location with predominantly negative responses, a discrepancy that may be associated with this cohort—especially given that the primary mode of transportation in The Villages is golf cart mobility. Golf carts in this community are capable of faster speeds than 10 mph, culturally accepted, and often available at little to no cost for residents. These advantages may currently overshadow the perceived benefits of AS, particularly given the limitations of AS, such as restricted routes, lower speeds,

and fixed schedules, which may not align with the flexible and convenient transportation expectations of this population. Across all locations, concerns about incidents or complaints post-implementation were raised, aligning with previous research as participants strongly indicated that accidents involving AS would discourage them from using such services [17]. Additionally, a preference for the freedom and control of self-driving was expressed, particularly among Lake Nona participants (younger to middle-aged adults). This aligns with a previous study where the majority of respondents declared a preference for shared control between humans and self-driving cars [37].

### 4.5. Experience with AV

For Experience with AV, the fifth most common theme, the dominance of positive frequency counts was consistent across all study locations. Participants shared insights into their past encounters with AVs, encompassing various experiences such as airport shuttles, Teslas, and vehicles equipped with Advanced Driver Assistance Systems like adaptive cruise control. The absence of negative narrative examples underscores the overall positive nature of participants' interactions with AVs, suggesting a favorable perception of this emerging technology. These findings are supported by literature indicating that positive exposure and experiences with AVs contribute to fostering trust, acceptance, and positive attitudes toward acceptance of autonomous transportation [17,37].

### 4.6. Perception Change

Perception Change, the last most common theme, offered insights into how participants' attitudes and perspectives changed before and after their experiences with the AS. Positive narratives from participants in Gainesville and Lake Nona showed increases in perceptions post-AS exposure. These individuals reported increased excitement, positive shifts in attitudes, and a growing comfort and understanding of AS technology. For those already favorably disposed towards AVs, the AS experience acted as a reinforcement or validation of their existing positive beliefs. Intriguingly, participants with initial reservations underwent a positive shift, suggesting that direct exposure to the AS contributed to an improvement in their perceptions. These findings align with previous research indicating an increase in perceptions after exposure to the AS [17].

Conversely, The Villages participants did not exhibit a positive attitude shift, indicating a unique dynamic in this location. Another study reported that older adults showed an increase in perceptions and expressed to be impressed by the AS' abilities after riding it [38]. As such, other factors than age may be responsible for the current perceptual state of the participants in the Villages. For example, the golf cart culture as described above, opens plausible opportunities for further research in this area. Overall, negative narratives across all sites predominantly revolved around unmet expectations, with participants expressing disappointment in aspects such as distance covered, smoothness of the ride, and overall experience. These perceptions underscore the ongoing need for technological advancements to meet Veterans' expectations and enhance the overall AS experience.

### 4.7. Co-Occuring Thematic Constructs

The co-occurring thematic constructs offer valuable insights into the intricate connections within participants' discussions on AS technology. Notably, the high co-occurrence of Perceived Benefits with Safety underscores participants' primary association of benefits with safety aspects, including the potential of AS to reduce accidents through minimizing human errors. This relationship suggests that safety is not merely a discrete theme but a key driver of perceived value—participants frequently viewed AS as beneficial precisely because of its potential to enhance road safety, particularly for vulnerable users such as older adults and Veterans with mobility impairments. The anticipated reduction in crash

risk, stress-free navigation, and mitigation of human error were interpreted as both direct safety gains and indirect quality-of-life improvements. As such, safety is not just an outcome but an important factor influencing participants' broader perception of benefit, where trust in the vehicle's ability to operate safely drives acceptance.

Moreover, the co-occurrence between Perceived Ease of Use and AS Experience highlights participants' emphasis on the convenience and user-friendliness of AS services. The high co-occurrence between Availability and AV Adoption indicates that participants link the availability of AS services with their perceptions of adopting AVs, suggesting that service routes and destinations influence adoption perceptions. The high co-occurrence between Accessibility and Safety reflects participants' awareness of safety considerations in the context of accessibility, particularly recognizing AS services as a safer alternative transportation for the elderly and people with disabilities. However, concerns were expressed about safety-related aspects, such as entering and exiting the vehicle, the need for safety operators, and ensuring compliance with the Americans with Disabilities Act (ADA).

The high co-occurrence between AS Experience and Safety, along with the subtheme Speed and Safety, highlights that safety considerations are highly intertwined with discussions about participants AS ride experience, particularly in relation to the vehicle's speed. The overall assessment of the AS Experience elicited a mix of positive and negative perceptions. Some participants were appreciative of the shuttle's meticulous safety measures and cautious operation, while others raised concerns about its slow speed and the potential for rear-end collisions. Notably, Safety, encompassing the subtheme Trust and reliability, demonstrated the highest co-occurrence with AS Experience, further underscoring the connection between these two themes.

Comfort, co-occurring with Perceived Benefits and AV Adoption, indicates that participants consider comfort a significant factor in evaluating the overall benefits and adoption of AS technology. In the context of Experience with AV and Perception Change, the high co-occurrence percentage with Safety underscores the impact of safety considerations on participants' experiences and how these experiences shape their perceptions of AS. Additionally, the co-occurrence of AV Adoption and External Variables with Perceived Benefits underscores participants' acknowledgment of perceived benefits influencing the adoption of AS technology, encompassing increased transportation to VA appointments and leisure activities, a relaxed ride, and potential cost savings. The co-occurrence patterns provide a nuanced understanding of how participants intertwine various themes and subthemes in their discussions about AS technology. These findings contribute to the existing literature on AS acceptance by highlighting the multifaceted nature of participants' considerations and the interconnectedness of key themes in shaping their perceptions and attitudes toward accepting (or not) the AS as a mode of transportation.

### 4.8. Integration with Existing Literature and Contributions

The focus groups, consisting of 26 participants across nine sessions in three different locations, provided valuable insights into participants' perceptions on AVs, allowing for a more in-depth understanding of the factors influencing their acceptance (or not) of AS technology. The focus group themes reveal a similarity to deductive codes identified in previous research on AS perceptions [17,18]. As demonstrated in Table 1, it is clear that the parallels between the deductive codes and the subthemes underscore the trustworthiness of the findings. Specifically, this synergy suggests that the qualitative findings of this study may have relevance in other research contexts, as they reflect consistency with existing literature on AS perceptions [17,18].

This study, focused on the Veteran population, offers a distinctive perspective on AS acceptance by contributing new insights that differentiate it from existing research on the

general population. Veterans' transportation needs, as identified in this study, are uniquely tied to their reliance on healthcare-related services, such as attending VA hospital appointments. This is in contrast with studies of the general population, where transportation needs are broader and less centered on healthcare access [39]. Additionally, this study revealed nuances in speed perceptions that were different from prior findings. While studies involving the general population often associate low AS speeds with increased safety perceptions [40], Veterans in this study raised concerns that slow speeds could pose safety risks, particularly in high-density traffic scenarios. Thus, particularly in urban or high-traffic areas, the balance between cautious operation and efficient performance is important to foster perceptions of safety. Moreover, prior research indicates that older participants typically demonstrate lower acceptance of AS technology [41–43]. However, this study expands this understanding by showing that among older Veterans in The Villages, local cultural norms, rather than age alone, may have also influenced perceptions—a nuance less frequently addressed among studies targeted to the general population studies.

Furthermore, Veterans in this study indicated the potential of AS services to enhance accessibility for vulnerable groups, such as individuals with disabilities or limited mobility. However, previous research has shown that respondents with special needs, particularly those with mobility restrictions, tend to perceive AVs negatively, citing concerns over safety and a general distrust of the technology [44]. In contrast, additional research demonstrates that exposure to AS can lead to positive experiences and increased acceptance among people with disabilities [14,15]. This contrast indicates the potential of hands-on AS experiences in shifting perceptions among mobility vulnerable groups.

To contextualize these findings, a comparison between AS and traditional transportation methods is important to better assess the utility of AS. Traditional transportation options available to Veterans—such as fixed-route public buses, paratransit, VA-provided vans, or personal vehicles—often provide greater route flexibility, faster speeds, and fewer constraints on scheduling. However, they also present notable barriers, including long wait times, limited service in rural or underserved areas, accessibility limitations, and the need for physical or cognitive effort to navigate the system. In contrast, participants perceived AS as potentially more predictable, relaxing, and accessible, particularly for those with mobility impairments or who were no longer driving. Yet, these benefits were sometimes diminished by the AS's slower speeds, limited coverage area, and current reliance on safety operators. The comparative strengths and limitations across modes suggest that AS may be best suited as a complementary solution—particularly for first-mile/last-mile service to VA facilities, or in low-demand areas—rather than a direct replacement for existing transportation infrastructure. However, a cost–benefit analysis and service efficiency evaluation will be necessary to clarify how AS compares operationally and economically to conventional services for this population.

In fact, one key limitation that emerged from this study—and that remains insufficiently addressed in both practice and research—is the issue of cost. Although participants noted the potential of AS to reduce transportation expenses, especially for long-distance travel to VA appointments, their feedback centered specifically on out-of-pocket costs for Veterans—that is, whether they would have to pay to access and use the AS service. However, the practical deployment of AS also demands consideration of system-level costs, such as those incurred by the VA or partnering agencies. These include capital investment, labor, maintenance, and technology infrastructure. For instance, factors such as route length, shuttle speed, frequency of service, and supervision model (on-board operator vs. remote monitoring) can significantly influence the overall cost of providing AS [45]. Remote supervision, while potentially reducing labor costs, introduces technical complexities and scaling challenges that must be carefully weighed. Current pilot programs remain

expensive and too limited in scale to guarantee cost-effectiveness—particularly if multiple vehicles or extensive mapping are required [45].

Lastly, this study complements and extend our research team's previously published work [46], which quantitatively, via a pre- and post-survey, assessed Veterans' perceptions before and after exposure to AS. Quantitative findings from the prior study revealed that pre-AS ride Total Acceptance median scores ranged between 65.8 and 69.6 (out of 100), indicating a moderately positive baseline perception. Participants' uniformly positive perceptions of their prior experiences with AVs (captured in this study under the qualitative theme of Experience with AV) may have contributed to these increased pre-ride Acceptance scores. This indicates the influence of previous exposure to AV technology on Acceptance. Additionally, the qualitative findings in this study—indicating positive frequency counts across themes such as Safety, Perceived Benefits, Experience with AV, AV Adoption, and Perception Change—substantiate the prior research's findings, which show an increase in Total Acceptance following AS exposure. Furthermore, the Perception Change theme is important as it indicated the influence of the AS experience, supported by both significant differences in Total Acceptance median scores and positive qualitative perceptions. Thus, by allowing individuals, including mobility vulnerable populations, to directly interact with the technology may contribute to driving positive perception changes toward AS acceptance.

## 5. Conclusions

This study assessed the lived experiences of Veterans post-AS exposure, leveraging focus group data from two studies. Positive perceptions were evident across themes such as Safety, Experience with AV, AV Adoption, Perceived Benefits, and Perception Change, despite negative perceptions regarding AS Experience. Safety, Perceived Benefits, and AS Experience emerged as dominant themes. While participants largely recognized the potential advantages of AS, concerns regarding operational speed, comfort, and technological reliability were noted. These findings indicate the need for continuous refinement of AS technology to align with user expectations. Additionally, participants expressed keen interest in AV Adoption, contingent on factors like convenience and expanded routes. Continuous collaboration with industry partners, further field testing, and overcoming AS technical and operational challenges (e.g., AS weather tolerance) are necessary steps in paving the way for considering AS services for Veterans.

This study contributes new insights to the literature by focusing specifically on Veterans—a population whose transportation needs are uniquely shaped by access to healthcare services. Unlike prior studies involving the general population, our findings suggest that Veterans may perceive slow AS speeds as unsafe in high-traffic settings, challenging assumptions that slower speeds universally enhance perceptions of safety. Additionally, among older Veterans in The Villages, local cultural norms appeared to influence acceptance of AS, highlighting that contextual factors beyond age may shape technology perceptions. Veterans also recognized the potential of AS to enhance accessibility for individuals with disabilities or limited mobility, contrasting with prior work suggesting skepticism from these groups. Further, while participants viewed AS as a promising complement to existing transportation modes, they emphasized the importance of route expansion, system reliability, and convenience—particularly to VA hospitals.

The implications of this research are significant for public health, policy, and industry. By addressing transportation as a critical social determinant of health, AS technology offers an avenue to reduce disparities in healthcare access, particularly for underserved Veterans. Policymakers and transportation planners are encouraged to prioritize safety enhancements, ensure ADA compliance, and develop tailored interventions to meet the unique needs of

diverse populations, including older adults and individuals with disabilities. Furthermore, industry stakeholders should focus on improving user experiences through features such as increased comfort, enhanced speed, and reliable performance, as well as fostering trust through safety measures like onboard operators. Overall, VA decision-makers (e.g., VA medical centers), VA transportation stakeholders (e.g., Veterans Transportation Service), and industry partners (e.g., AS manufacturers) can leverage these findings to educate, advocate, and address concerns identified in the study's findings. By integrating Veterans' perspectives into the design and implementation of AS programs, decision-makers may facilitate the adoption of AS as a reliable and equitable transportation option for Veterans.

*5.1. Limitations*

Limitations in focus groups include their reliance on Teams videoconferencing, potentially excluding participants without access to advanced technologies and encountering technical difficulties. Additionally, self-selection bias might have influenced the findings, as the perspectives shared in the focus groups could be skewed by the characteristics and culture of those who chose to participate. Although our participants included individuals across different military categories, ethnic minorities, and people with disabilities, these demographic details were not consistently collected across all study sites. As a result, the research team decided not to report these characteristics in the manuscript. Participants' English proficiency was assessed based on self-reported fluency. This approach was used to ensure that individuals could understand the focus group questions and effectively engage in the discussion. While we acknowledge that this method may introduce variability in actual language proficiency, no issues related to language comprehension or communication were identified by the research team during data collection.

Notably, the majority of participants were from urban areas, which may limit the generalizability of findings to rural communities. Our findings should therefore be interpreted primarily through the lens of urban and suburban contexts. Among the nine focus group sessions, two had fewer than three participants, which may have influenced group dynamics and limited the depth of discussion. Acknowledging this limitation, the research team chose to retain these sessions, as their findings were consistent with those from larger groups [47]. Despite this challenge, the study yielded rich qualitative data that enhanced understanding of participant attitudes toward AS technology.

While the study employed credibility strategies such as investigator triangulation to ensure trustworthiness, certain aspects of trustworthiness were not fully addressed. Specifically, to enhance the dependability of qualitative findings, establishing an audit trail or ensuring stepwise replication of data could have been beneficial, allowing for a detailed track record of the data collection process and measurement of coding accuracy and inter-coders' reliability. Furthermore, to improve transferability, quantifying operational data saturation could have been used to extend the degree to which the results can be generalized or transferred to other contexts or settings.

The predominantly White composition of the research team and the absence of a Veteran as a research personnel member in the data collection process, may have introduced bias into various aspects of the study, including its design, data collection, and analysis. This limitation can manifest in subtle ways, such as framing interview questions or interpreting participant responses through a lens that may not adequately capture the nuances of the Veteran experience. Furthermore, the absence of a Veteran as a research personnel member in the data collection process may hinder the team's ability to establish rapport and trust with Veteran participants. Veterans may feel more comfortable sharing their experiences and insights with someone who has firsthand knowledge of military life and the challenges

they face. Without this insider perspective, the research team may inadvertently overlook important details or fail to ask probing questions that could yield richer qualitative data.

*5.2. Strengths and Future Directions*

Integrating AS into community mobility systems presents a promising technological solution, particularly for Veterans who face challenges with driving or choose not to drive. This study stands out as one of the first to examine the lived experiences of Veterans with AS technology in the United States. By focusing on three geographically diverse locations in Florida, the research broadens its relevance to varying regional contexts and mobility needs. The study leverages innovative technology while fostering collaboration with key transportation stakeholders and industry leaders to gather insights into how Veterans perceive and engage with AS. Through qualitative focus group discussions, the research captures detailed narratives that reveal the nuanced connections between individual experiences and the wider acceptance of AVs. This approach deepens our understanding of Veterans' attitudes and perceptions toward AS, lays a foundation for future studies, and informs decision-makers, transportation stakeholders, and industry professionals in developing AS programs tailored to Veterans' needs.

For industry partners and stakeholders in autonomous transportation and community mobility, this study provides insights into improving user experiences and acceptance of AVs, specifically among Veterans. Key thematic constructs such as Safety, Comfort, and AV Adoption offer focal points for intervention. Addressing concerns related to AS experience, operational speed, and safety can greatly enhance user satisfaction. Stakeholders may consider age-related factors influencing acceptance, particularly with older Veterans expressing more safety concerns. Introducing safety operators onboard may heighten user confidence and greater sense of safety. Furthermore, stakeholders may recognize the potential benefits of AS technology for specific demographics, e.g., older adults and people with disabilities, emphasizing accessibility and safety features. Continuous collaboration with industry partners, testing, and overcoming operational challenges are essential steps in paving the way for considering AS adoption for Veterans.

Notably, this study did not collect or analyze data on the projected costs of implementing AS within the VA system itself (e.g., infrastructure investment, staffing, maintenance), which constitutes a meaningful gap. Understanding system-level cost implications—from the perspective of the VA or other public agencies—is needed for evaluating the feasibility and scalability of integrating AS into healthcare transportation networks. Future research may examine both sides of the cost equation: the affordability of AS from the end-user perspective and the financial viability from a service-provider or policy standpoint.

Findings from this study provide insights into how Veterans' acceptance of AS may serve as a catalyst for considering AS for Veterans. This could improve transportation options, ultimately reducing healthcare disparities associated with transportation barriers and fostering Veterans' community integration and adjustment to civilian life. Moreover, transportation-related accidents, often stemming from human error, pose substantial public health risks. Autonomous shuttle technology has the potential to mitigate these risks by minimizing human error, thereby decreasing the incidence of accidents and associated injuries among Veterans and the general population. These findings also carry important policy implications. Policymakers may prioritize addressing safety concerns, comfort, and operational speed to foster widespread AS acceptance. Moreover, recognizing regional variations in perceptions is essential, necessitating tailored policies to accommodate diverse geographical contexts. By developing targeted interventions, policymakers can enhance acceptance levels among specific populations, such as older adults, facilitating a more effective integration of AS into diverse communities.

The study has important public health implications, particularly in addressing transportation as a social determinant of health. Transportation barriers not only delay Veterans' access to critical healthcare services, resulting in poorer health outcomes, but also limit their participation in employment, social activities, and community engagement. These challenges often exacerbate feelings of isolation, particularly among Veterans at higher risk of mental health conditions such as depression and PTSD, further diminishing their quality of life. The introduction of AS technology, especially in rural areas where distances to healthcare facilities are substantial, offers valuable benefits to Veterans. However, to better inform the implementation of AS technology across all Veteran populations, future research should prioritize the inclusion of rural Veterans to address existing gaps in the literature and assess unique transportation needs in geographically isolated settings.

**Author Contributions:** Conceptualization, S.C.; methodology, S.C.; formal analysis, I.W., S.W.H. and N.E.S.; investigation, I.W., K.L. and N.E.S.; resources, S.C.; data curation, I.W.; writing—original draft preparation, I.W.; writing—review and editing, S.C., N.E.S., S.W.H. and K.L.; visualization, I.W.; supervision, S.C. and N.E.S.; project administration, S.C. and N.E.S.; funding acquisition, S.C. All authors have read and agreed to the published version of the manuscript.

**Funding:** This research was funded by the VA Office of Rural Health, project number P0213747.

**Institutional Review Board Statement:** The study was conducted in accordance with the Declaration of Helsinki, and approved by the Institutional Review Board of the University of Florida (FY21-22 IRB202101463: PI Classen and FY22-23 IRB202202386: PI Classen).

**Informed Consent Statement:** Informed consent was obtained from all subjects involved in the study.

**Data Availability Statement:** The original contributions presented in the study are included in the article; further inquiries can be directed to the corresponding authors.

**Acknowledgments:** The research acknowledges the support provided by the University of Florida's Institute for Driving, Activity, Participation & Technology, under the leadership of Sherrilene Classen, Principal Investigator of the two studies. Additionally, the collaboration and resources of the North Florida/South Georgia Veterans Health System, the Malcom Randall VA Medical Center, and the Gainesville Office of Rural Health were pivotal to the successful execution of this study.

**Conflicts of Interest:** The authors declare no conflicts of interest. The funder had no role in the design of the study, in the collection, analyses, or interpretation of data, in the writing of the manuscript, or in the decision to publish the results.

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
