# Peer review of "Assessing Veterans’ Lived Experiences After Exposure to an Autonomous Shuttle"

_futuretransp, doi:10.3390/futuretransp5030095_

Round 1
Reviewer 1 Report
Comments and Suggestions for Authors
This manuscript presents a qualitative study on the acceptance of autonomous shuttles (AS) among veterans. The results indicate that most participants held positive perceptions, with safety, perceived benefits, and AS experience emerging as dominant themes. The research content is of interest and well-structured. However, there are a few concerns that need to be answered:
(1) The cost aspect is not addressed, which constitutes a significant limitation. Cost is a crucial factor in deploying AS for veteran healthcare access and should be included to enhance the practicality of the study.
(2) The study lacks a comparison with traditional transportation methods. Such a contrast is essential to fully assess the pros and cons of autonomous shuttles. The authors should add a detailed comparison to deepen and validate their findings.
(3) Given that factors like perceived benefits and safety are interrelated, the interrelationships among these factors should also be further elucidated.
(4) It is recommended to enhance the readability of the article by including visual illustrations in the results comparison section.
Author Response
|
1. Summary |
|
|
|
Thank you very much for taking the time to review this manuscript. Please find the detailed responses below and the corresponding revisions in track changes in the re-submitted files.
|
||
|
2. Point-by-point response to Comments and Suggestions for Authors |
||
|
Comments 1: The cost aspect is not addressed, which constitutes a significant limitation. Cost is a crucial factor in deploying AS for veteran healthcare access and should be included to enhance the practicality of the study. |
||
|
Response 1: Thank you for your feedback. We have revised the introduction (Pages 2-3, Lines 91–98), discussion (Page 21, Lines 787–800), and conclusion (Page 24, Lines 954–961) sections to acknowledge the cost of deploying AS.
|
||
|
Comments 2: The study lacks a comparison with traditional transportation methods. Such a contrast is essential to fully assess the pros and cons of autonomous shuttles. The authors should add a detailed comparison to deepen and validate their findings. |
||
|
Response 2: Thank you for your comment. We have revised the manuscript to add a comparison with traditional transportation methods to better assess the pros and cons of AS (Page 21, Lines 771–786).
Comments 3: Given that factors like perceived benefits and safety are interrelated, the interrelationships among these factors should also be further elucidated. Response 3: Thank you for your feedback. We have revised the discussion section (Page 19, Lines 692–699) to further elucidate the interrelationship between perceived benefits and safety. |
||
Comments 4: It is recommended to enhance the readability of the article by including visual illustrations in the results comparison section.
Response 4: Thank you for your suggestion. We have added a visual illustration in the results section (Page 16, Lines 535–538).
Reviewer 2 Report
Comments and Suggestions for Authors
- Abstract section Transportation is often cited by Veterans, including minority groups, people with disabilities, and residents of rural areas. Does this sentence mean minority groups, people with disabilities, and residents of rural areas included among veterans? The description is ambiguous and fails to understand the author's true meaningï¼›
- 2.2 The Study Design is not complete enough and needs to be supplementedï¼›
- 2.3 The sentence "Study Population, Veterans were eligible regardless of combat history or service branch" does not quite match the content. How is fluent in English judged? Are there any relevant standards?
- Can the number of respondents in the text support the research of this article?
- 3.5 This part of AV Adoption doesn't seem to have an academic style;
- 3.7 co-occuring Thematic Constructs The first paragraph is missing the final punctuation mark;
- It is suggested that the author present the new findings of the research in the conclusion section of the article, highlighting the research content and key points.
The language of this article still needs further revision and polishing.
Author Response
|
1. Summary |
|
|
|
Thank you very much for taking the time to review this manuscript. Please find the detailed responses below and the corresponding revisions in track changes in the re-submitted files.
|
||
|
2. Point-by-point response to Comments and Suggestions for Authors |
||
|
Comments 1: Abstract section Transportation is often cited by Veterans, including minority groups, people with disabilities, and residents of rural areas. Does this sentence mean minority groups, people with disabilities, and residents of rural areas included among veterans? The description is ambiguous and fails to understand the author's true meaning. |
||
|
Response 1: Thank you for your comment. We have revised the abstract accordingly (Page 1, Lines 11–12). |
||
|
Comments 2: 2.2 The Study Design is not complete enough and needs to be supplemented. |
||
|
Response 2: Thank you for your feedback. We have revised the study design (Page 4, Lines 161–167).
Comments 3: 2.3 The sentence "Study Population, Veterans were eligible regardless of combat history or service branch" does not quite match the content. How is fluent in English judged? Are there any relevant standards? Response 3: Thank you for your comment. We are unclear on how the sentence does not match the content but are happy to make further changes with clarification. We have revised the section to clarify how English fluency was determined (Page 4, Lines 169–171).
Comments 4: Can the number of respondents in the text support the research of this article? Response 4: Thank you for your question. We have clarified in the Discussion section (Page 16, Lines 542–546) that the number of respondents aligns with qualitative research standards. Additionally, we have indicated in the limitations section (Page 23, Lines 913–915) that “to improve transferability, quantifying operational data saturation could have been used to extend the degree to which the results can be generalized or transferred to other contexts or settings.” |
||
Comments 5: 3.5 This part of AV Adoption doesn't seem to have an academic style.
Response 5: Thank you for your comment. We were unsure what specifically was meant by “academic style” in this context, but we are happy to revise the section further upon clarification.
Comments 6: 3.7 co-occuring Thematic Constructs The first paragraph is missing the final punctuation mark.
Response 6: Thank you for your comment. We have corrected the punctuation at the end of the first paragraph in the Co-occurring Thematic Constructs section (Page 13, Line 505).
Comments 7: It is suggested that the author present the new findings of the research in the conclusion section of the article, highlighting the research content and key points.
Response 7: Thank you for the suggestion. We have revised the conclusion to more clearly highlight the key findings and contributions of the research (Page 22, Lines 863–874).
Reviewer 3 Report
Comments and Suggestions for Authors
The paper is about a topic rarely addressed in international transportation literature. Veterans constitute a special group in many aspects, including transportation.
The size of the group in the survey is quite limited, but the survey design, implementation, analysis of the data as well as the discussion are of high quality.
Although I feel that the paper with its 24 pages is a little bit overwritten, I have no definite recommendation for cuts.
Author Response
|
1. Summary |
|
|
|
Thank you very much for taking the time to review this manuscript. Please find the detailed responses below.
|
||
|
2. Point-by-point response to Comments and Suggestions for Authors |
||
|
Comments 1: The paper is about a topic rarely addressed in international transportation literature. Veterans constitute a special group in many aspects, including transportation. The size of the group in the survey is quite limited, but the survey design, implementation, analysis of the data as well as the discussion are of high quality. Although I feel that the paper with its 24 pages is a little bit overwritten, I have no definite recommendation for cuts. |
||
|
Response 1: Thank you for your feedback. We appreciate your recognition of the study’s contributions and quality. |
||
Reviewer 4 Report
Comments and Suggestions for Authors
I applaud the approach to explore Veterans’ experiences with autonomous shuttles. It is an important topic and directed content analysis was a suitable choice. The limitations of using focus groups are clearly explained. Overall, you have documented your efforts thoroughly and effectively.
I have one suggestion to further improve the clarity and interpretation of your results:
- Consider adding a table summarizing the demographic information of participants at the beginning of the Results section. Accompany this with a brief summary in the text about your purposive sampling. I recommend including at least the following details: age, sex, race/ethnicity (if collected), disability status or mobility vulnerability (e.g., musculoskeletal or neurological conditions, or combat-related injuries such as amputations, spinal cord injuries, or traumatic brain injuries). This context would reveal more interpretable findings.
Otherwise, excellent work!
Author Response
|
1. Summary |
|
|
|
Thank you very much for taking the time to review this manuscript. Please find the detailed responses below.
|
||
|
2. Point-by-point response to Comments and Suggestions for Authors |
||
|
Comments 1: I applaud the approach to explore Veterans' experiences with autonomous shuttles. It is an important topic and directed content analysis was a suitable choice. The limitations of using focus groups are clearly explained. Overall, you have documented your efforts thoroughly and effectively. I have one suggestion to further improve the clarity and interpretation of your results: Consider adding a table summarizing the demographic information of participants at the beginning of the Results section. Accompany this with a brief summary in the text about your purposive sampling. I recommend including at least the following details: age, sex, race/ethnicity (if collected), disability status or mobility vulnerability (e.g., musculoskeletal or neurological conditions, or combat-related injuries such as amputations, spinal cord injuries, or traumatic brain injuries). This context would reveal more interpretable findings. Otherwise, excellent work! |
||
|
Response 1: Thank you for your feedback and suggestion to include a table summarizing demographic characteristics. We agree that such contextual information can enhance interpretability. Unfortunately, detailed demographic information such as race/ethnicity, or specific disability and mobility-related conditions (e.g., musculoskeletal or neurological injuries) was not collected. As a result, we provided a descriptive summary (instead of a table) of the demographic information. |
||
Round 2
Reviewer 1 Report
Comments and Suggestions for Authors
The authors have addressed all my comments. So I recommend the manuscript to be accepted for publication
Author Response
|
Comments 1: The authors have addressed all my comments. So I recommend the manuscript to be accepted for publication. |
|
Response 1: Thank you very much for taking the time to review this manuscript. |
Reviewer 2 Report
Comments and Suggestions for Authors
- Regarding the answer to the first question, I now want to know which category of military respondents you specifically investigated in your article are from ethnic minorities, the disabled, or those living in rural areas.Are there any disabled people among the respondents?
- Regarding the answer to question 2, I would like to thank the author for making the revision. The author failed to understand my meaning. What I hope to see is the design of the research approach and process throughout the entire article, not the design of the investigation part. I suggest the author make additional comments.
- In response to question three, the author doesn't understand the confusion I raised. Now, I'll explain it in a different way. I hope the author can explain the meaning of the following sentence. "Veterans were eligible regardless of combat history or ser vice branch." The author's standard requirement for respondents' English proficiency is "self-reported as fluent in. "English" does not have academic rigor. It is suggested to choose other methods to replace this one.
- Regarding the answer to question 4, I couldn't find the corresponding text mentioned by the author. I suggest the author recheck the page numbers of the text. There is no such number as page 16. It is recommended to make revisions and resubmit.
- Regarding Suggestion 5, the author believes he has not understood it. I will restate my viewpoint. I don't think this part of 3.5 AV Adoption should appear in scientific research papers. I hope the author can understand my meaning.
Author Response
|
1. Summary |
|
|
|
Thank you very much for taking the time to review this manuscript. Please find the detailed responses below and the corresponding revisions in track changes in the re-submitted files.
|
||
|
2. Point-by-point response to Comments and Suggestions for Authors |
||
|
Comments 1: Regarding the answer to the first question, I now want to know which category of military respondents you specifically investigated in your article are from ethnic minorities, the disabled, or those living in rural areas. Are there any disabled people among the respondents? |
||
|
Response 1: Thank you for your question. Although our participants included individuals across different categories of the military, ethnic minorities, and disability status, these demographic details were not consistently collected across the different study locations. Therefore, the research team decided not to report them in the manuscript. We have clarified this decision in the methods section (Page 5, Lines 246–249) and limitations section of the paper (Page 23, Lines 920–923). Regarding urban vs rural areas, this information has been reported on Page 6, Line 284. |
||
|
Comments 2: Regarding the answer to question 2, I would like to thank the author for making the revision. The author failed to understand my meaning. What I hope to see is the design of the research approach and process throughout the entire article, not the design of the investigation part. I suggest the author make additional comments. |
||
|
Response 2: Thank you for your feedback and for clarifying your original point. We appreciate your suggestion and are happy to make any additional changes or provide further explanation to strengthen the manuscript. To ensure we address your concerns thoroughly, could you please clarify which specific aspects of the research approach and process you feel are currently unclear or missing? We want to ensure that the study’s overall design—from its conceptual foundation through execution and interpretation—is clearly communicated throughout the manuscript.
Comments 3: In response to question three, the author doesn't understand the confusion I raised. Now, I'll explain it in a different way. I hope the author can explain the meaning of the following sentence. "Veterans were eligible regardless of combat history or ser vice branch." The author's standard requirement for respondents' English proficiency is "self-reported as fluent in. "English" does not have academic rigor. It is suggested to choose other methods to replace this one. Response 3: Thank you for the clarification. Regarding the sentence “Veterans were eligible regardless of combat history or service branch,” this was meant to indicate that eligibility for participation in the study was not limited by military service experience, including whether or not the Veteran had served in combat or in a particular military branch (e.g., Army, Navy, Air Force, etc.). We have revised this sentence for clarity (Page 4, Lines 173–176). We acknowledge your concern regarding the phrase “self-reported as fluent in English.” This phrasing was used as part of the inclusion criteria for participants, consistent with qualitative research practices where language proficiency is commonly determined through self-report. However, we understand the need for greater academic rigor and will clarify this in the manuscript by elaborating on why self-reported fluency was appropriate in this context (Page 6, Lines 169–173). We have also noted this as a limitation (Page 23, Lines 923–928).
Comments 4: Regarding the answer to question 4, I couldn't find the corresponding text mentioned by the author. I suggest the author recheck the page numbers of the text. There is no such number as page 16. It is recommended to make revisions and resubmit. Response 4: Regarding the reviewer’s previous question on the number of respondents, we have clarified in the Discussion section (Page 16, Lines 563–567) that the number of respondents aligns with qualitative research standards. Additionally, we have indicated in the limitations section (Page 23, Lines 942–944) that “to improve transferability, quantifying operational data saturation could have been used to extend the degree to which the results can be generalized or transferred to other contexts or settings.” |
||
Comments 5: Regarding Suggestion 5, the author believes he has not understood it. I will restate my viewpoint. I don't think this part of 3.5 AV Adoption should appear in scientific research papers. I hope the author can understand my meaning.
Response 5: Thank you for the clarification. We understand your concern regarding the inclusion of Section 3.5 “AV Adoption.” However, we respectfully note that this content represents one of the key emergent themes from our qualitative data and reflects participants’ perceptions and discussions around the adoption of AV technology. As such, it is important to report this theme in the manuscript to accurately represent the data and maintain the integrity of the thematic analysis. That said, we have reviewed the section to ensure that it is presented in a scholarly and evidence-based manner, consistent with the expectations of a scientific research paper. We are open to revising the framing or tone of this section further if you have specific suggestions.
Round 3
Reviewer 2 Report
Comments and Suggestions for Authors
The author has made two rounds of revisions, but the quality of the article has not improved significantly. I think it does not meet the acceptance requirements at present. It is suggested that the author be more careful when revising academic papers in the future, and carefully check the page numbers and line numbers of the revised parts when writing reply opinions.
- Regarding the answer to the author's first question, I did not find the modification marks on Page 5, Lines 246-249 and Page 23, Lines 920-923. Moreover, the content of Page 6, Line 284 has neither the modification marks nor the so-called urban and rural related information of the author in the article.
- Regarding Suggestion Two, I hope to see a design of the overall method and process of the article, and I hope the author can understand it.
- Regarding Suggestion 5, it is recommended that the author place this part in the appendix and try not to include it in the main body of the paper.
Author Response
|
Response to Reviewer 2 Comments
|
||
|
1. Summary |
|
|
|
Thank you very much for taking the time to review this manuscript. Please find the detailed responses below and the corresponding revisions in track changes in the re-submitted files.
|
||
|
2. Point-by-point response to Comments and Suggestions for Authors |
||
|
Comments 1: The author has made two rounds of revisions, but the quality of the article has not improved significantly. I think it does not meet the acceptance requirements at present. It is suggested that the author be more careful when revising academic papers in the future, and carefully check the page numbers and line numbers of the revised parts when writing reply opinions.
|
||
|
Response 1: Thank you for your continued time and effort in reviewing our manuscript. We apologize for the incorrect Page and Lines references. Please see the correct Page/Line information below. Although our participants included individuals across different categories of the military, ethnic minorities, and disability status, these demographic details were not consistently collected across the different study locations. Therefore, the research team decided not to report them in the manuscript. We have clarified this decision in the methods section (Page 4, Lines 205–208) and limitations section of the paper (Page 23, Lines 920–923). Regarding urban vs rural areas, this information has been already reported on Page 5, Line 284. |
||
|
Comments 2: Regarding Suggestion Two, I hope to see a design of the overall method and process of the article, and I hope the author can understand it. |
||
|
Response 2: We respectfully disagree with the assertion that the overall research design and process are unclear. The manuscript includes a detailed account of the study’s rationale, methodology, procedures, and analysis.
In our prior response, we specifically requested clarification on which aspects of the research approach were perceived as insufficient, but the feedback remained general. Without more targeted suggestions, we have done our best to ensure that the manuscript clearly communicates the study’s full design and process throughout. |
||
Comments 3: Regarding Suggestion 5, it is recommended that the author place this part in the appendix and try not to include it in the main body of the paper.
Response 3: We respectfully disagree with the recommendation to remove or relocate Section 3.5 “AV Adoption” to the appendix. This section presents a key theme directly drawn from participants' narratives and reflects their perceptions of and attitudes toward adopting AV technology. As this study uses a qualitative design, it is important to report all core themes that arose from the data.
We have revised the section to ensure that it aligns with academic standards in tone and structure, and we believe its inclusion in the main body of the paper is both appropriate and necessary for a complete presentation of findings.
We appreciate the reviewer’s viewpoint and are open to further refinement of the section’s wording if there are specific concerns regarding style or tone.